# Predicting cardiovascular disease risk using photoplethysmography and deep learning

**Wei-Hung Weng** [1‡*], **Sebastien Baur** [1‡], **Mayank Daswani** [1‡], **Christina Chen** [1], **Lauren Harrell** [1], **Sujay Kakarmath** [1], **Mariam Jabara** [1], **Babak Behsaz** [1], **Cory Y. McLean** [1], **Yossi Matias** [1], **Greg S. Corrado** [1], **Shravya Shetty** [1], **Shruthi Prabhakara** [1*], **Yun Liu** [1], **Goodarz Danaei** [2], **Diego Ardila** [1*]

1 Google LLC, Mountain View, California, United States of America, 2 Department of Global Health and Population, Department of Epidemiology, Harvard School of Public Health, Boston, Massachusetts, United States of America

‡ WHW, SB and MD contributed equally as first authors to this work.
* ckbjimmy@google.com (WHW); shruthip@google.com (SP); ardila@google.com (DA)

**Data Availability Statement:** This research has been conducted using the UK Biobank Resource under Application Number 65275. Individual-level data from the UK Biobank are not publicly available

## Abstract

Cardiovascular diseases (CVDs) are responsible for a large proportion of premature deaths in low- and middle-income countries. Early CVD detection and intervention is critical in these populations, yet many existing CVD risk scores require a physical examination or lab measurements, which can be challenging in such health systems due to limited accessibility. We investigated the potential to use photoplethysmography (PPG), a sensing technology available on most smartphones that can potentially enable large-scale screening at low cost, for CVD risk prediction. We developed a deep learning PPG-based CVD risk score (DLS) to predict the probability of having major adverse cardiovascular events (MACE: non-fatal myocardial infarction, stroke, and cardiovascular death) within ten years, given only age, sex, smoking status and PPG as predictors. We compare the DLS with the office-based refit-WHO score, which adopts the shared predictors from WHO and Globorisk scores (age, sex, smoking status, height, weight and systolic blood pressure) but refitted on the UK Biobank (UKB) cohort. All models were trained on a development dataset (141,509 participants) and evaluated on a geographically separate test (54,856 participants) dataset, both from UKB. DLS's C-statistic (71.1%, 95% CI 69.9–72.4) is non-inferior to office-based refit-WHO score (70.9%, 95% CI 69.7–72.2; non-inferiority margin of 2.5%, p<0.01) in the test dataset. The calibration of the DLS is satisfactory, with a 1.8% mean absolute calibration error. Adding DLS features to the office-based score increases the C-statistic by 1.0% (95% CI 0.6–1.4). DLS predicts ten-year MACE risk comparable with the office-based refit-WHO score. Interpretability analyses suggest that the DLS-extracted features are related to PPG waveform morphology and are independent of heart rate. Our study provides a proof-of-concept and suggests the potential of a PPG-based approach strategies for community-based primary prevention in resource-limited regions.

## Introduction

Cardiovascular diseases (CVDs) are responsible for one third of deaths globally [1] with approximately three quarters occurring in low- and middle-income countries (LMICs) where

according to the policy but will be made available after the application of UK Biobank. Please visit the UK Biobank website, https://www.ukbiobank.ac.uk/ , for application procedures. Code for collecting PPG in an app is available at https://github.com/ google-research/CVD-paper-mobile-camera- example. Statistical code used for this study will be available at https://github.com/Google-Health/ google-health. Embeddings for the UK Biobank PPG data will be returned to and made available via the UK Biobank.

**Funding:** The study was supported by Google LLC. All Google-affiliated authors are Google employees and own Alphabet stock. Google LLC was involved in the design and conduct of the study; analysis, and interpretation of the data; preparation, review, or approval of the manuscript; and decision to submit the manuscript for publication.

**Competing interests:** Author WHW, SB, MD, CC, SK, YL, and DA are employed at Google LLC and hold shares in Alphabet, and are co-inventors on patents (in various stages) for CVD risk prediction using deep learning and PPG, but declare no non-financial competing interests. LH, BB, CYM, YM, GSC, SS, SP are employed at Google LLC and hold shares in Alphabet but declare no non-financial competing interests. SK serves as an Associate Editor for this journal but had no role to play in the editorial process and decisions for this manuscript. GD declares no financial or non-financial competing interests.

there's a paucity of resources for early disease detection [2, 3]. Because CVD risk factors such as hypertension, diabetes, or hyperlipidemia are typically symptomless before advanced disease, there is a great need for screening programs to identify those at high risk of CVD events. Interventions such as lifestyle counseling, with or without prescription medications, have shown to be an effective strategy for CVD prevention among these individuals [4].

Multiple risk scores, such as WHO/ISH risk chart and Globorisk scores, have been developed to triage CVD risk based on demographics, past medical history, vital signs, and laboratory data [4–7]. However, the dependency of these risk scores on medical and laboratory equipment (e.g., sphygmomanometers) [8, 9] limits their reach. Specifically, low-resource healthcare systems have relied largely on opportunistic screening [10], such as via community healthcare workers (CHWs) [11], to close access gaps. We reasoned that developing low-cost, easy-to-use, lightweight, digital point-of-care tools using sensors already available in smartphones [12–14], could potentially further the reach and capability of CHW-based programs and enable large-scale screening at low cost [15].

Among sensing signals for the circulatory system, photoplethysmography (PPG) is a non-invasive, fast, simple, and low-cost technology, and can be captured with sensors available on increasingly ubiquitous devices such as smartphones and pulse oximeters [16]. PPG measures the change in blood volume in an area of tissue across cardiac cycles and is primarily used for heart rate monitoring in healthcare settings [17, 18]. Research has also investigated the utility of PPG in understanding short term fluctuations in vascular compliance, by estimating continuous blood pressure (BP) in an ICU setting [17, 19, 20], though the accuracy of such approaches is known to be insufficient even when per-user calibration is available [17]. Beyond short term vascular changes, research has also been conducted into understanding the slow manifestation of vascular aging and arterial stiffness from PPG waveforms [21–23], which are useful for longer-term CVD risk assessment. Since PPG is potentially more accessible and requires less training for measurement, such technologies could provide accurate real-time insights. The ubiquity of smartphones have also prompted research involving PPG as measured from smartphone cameras, via placing a finger on the camera [16]. Taken together, enabling CVD-risk estimation based on PPG signals can potentially be a highly accessible screening tool in low-resource health systems (Fig 1).

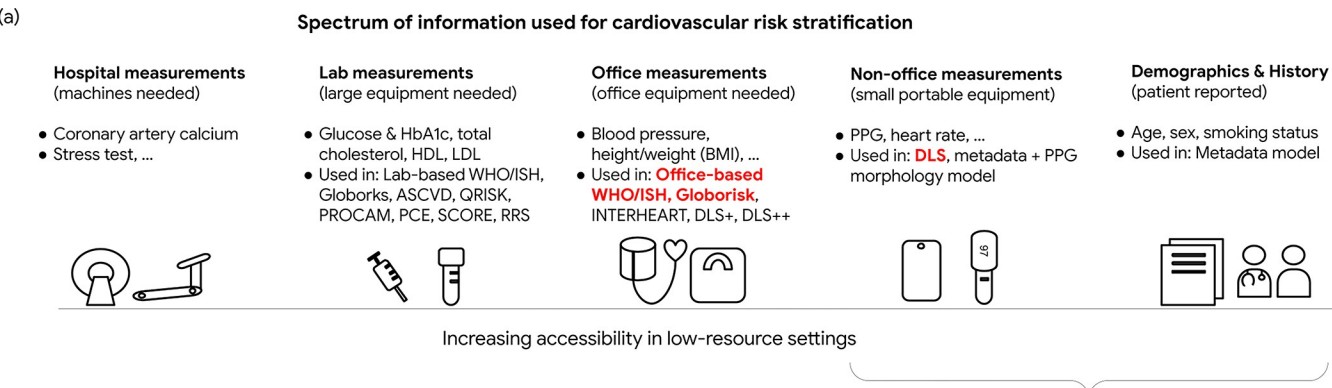

(a)

**Spectrum of information used for cardiovascular risk stratification**

**Hospital measurements**
(machines needed)
- Coronary artery calcium
- Stress test, ...

**Lab measurements**
(large equipment needed)
- Glucose & HbA1c, total cholesterol, HDL, LDL
- Used in: Lab-based WHO/ISH, Globorks, ASCVD, QRISK, PROCAM, PCE, SCORE, RRS

**Office measurements**
(office equipment needed)
- Blood pressure, height/weight (BMI), ...
- Used in: **Office-based WHO/ISH, Globorisk**, INTERHEART, DLS+, DLS++

**Non-office measurements**
(small portable equipment)
- PPG, heart rate, ...
- Used in: **DLS**, metadata + PPG morphology model

**Demographics & History**
(patient reported)
- Age, sex, smoking status
- Used in: Metadata model

Increasing accessibility in low-resource settings

Goal of our proposed method (DLS):
use easy-to-collection information, and equipment that
is already widely available in the community

**Fig 1. Summary of study motivation and design.** The motivation of applying the PPG-based cardiovascular disease (CVD) risk assessment in the low-resource health systems. Non-office based information acquired from mobile-sensing technologies may help address the burden of cardiovascular disease risk screening and triage in resource-limited areas. In this study, we compare our developed model (DLS) with the existing office-based and lab-based CVD risk scores that have been developed for low-resource medical settings, including models refitting variables in the WHO and Globorisk scores, and office-based and lab-based Globorisk scores recalibrated on the same study cohort.

While there are existing works on the BP estimation and evaluating other related CVD risk-factors such as heart rate variability (HRV) or arterial stiffness from PPG [24, 25], we have not found existing literature on predicting CVD risk directly from PPG waveform signals. The closest related works are those who predicted CVD risk via arterial stiffness index (ASI) estimated by PPG [26–28]. In this paper, we investigate the feasibility of leveraging PPG for CVD risk prediction using data from the UK Biobank (UKB). Specifically, we predict the ten-year risk of developing a major adverse cardiovascular event (MACE) using deep learning-based PPG embeddings and heart rate (measured by PPG), along with other demographics, including age, sex and smoking status, but without any inputs from physical examination or laboratory data (S1 Fig). We find that our deep learning PPG-based CVD risk prediction score (DLS) is well-calibrated and non-inferior to the existing comparative office-based CVD risk score using predictors from WHO/ISH and Globorisk, that requires blood pressure, weight and height measurement, or laboratory data.

## Material and methods

### Overview

We developed a new CVD risk prediction score, DLS, using age, sex, smoking status and the results of analysis of PPG signals using deep learning. We used a Cox proportional hazard model and data from UKB to predict the ten-year risk of MACE among individuals free of CVD at baseline.

### Data source and cohort

The DLS was developed and evaluated using data from the UKB dataset, filtered to focus on participants aged 40–74 to mirror a previous study [4]. We then stratified UKB participants who had PPG waveforms recorded into three subsets: train (n = 105,319), tune (n = 46,868), and test (n = 57,702) subsets based on geographic information on the site of data collection, i.e., latitude and longitude. This strategy aligns with TRIPOD guidelines [29] on external validation (specifically validation on a different geographic region) by allowing for non-random variation between data splits such as differences in data acquisition or environment.

We used PPG waveforms from all visits for the participants in this train subset to train the PPG feature extractor in DLS (details in "Model Development"). The low-dimensional numeric outputs (embeddings) computed by this model were used as additional input features to our Cox model. To develop the Cox model that generates DLS to predict MACE risk, additional clinical and demographic variables and inclusion/exclusion criteria were needed. First, we excluded participants with non-fatal myocardial infarction or stroke before their first visit, or missing any of variables for our model (age, sex, and smoking status). We also excluded those without body mass index (BMI) or systolic BP (SBP) for a fair comparison against the other office- and lab-based risk prediction models. For each participant, we only included measurements related to their first visit. All numerically measured variables were standard-scaled. Cox models were regularized using a ridge penalty. In the final cohort, 97,970, 43,539, and 54,856 participants were included to train, tune, and test the survival model, respectively (Fig 2). The participants were recruited to the UKB study between 2006 and 2010, and the anonymous individual participants' data in UKB was collected and accessible by authors, for modeling and data analysis from September 2022 to March 2023. Descriptive statistics for this cohort are in Table 1.

### Model development

**First stage: PPG feature extractor.**   For DLS, we first trained a deep learning-based feature extractor to learn PPG representations from UKB PPG summarized waveform signals,

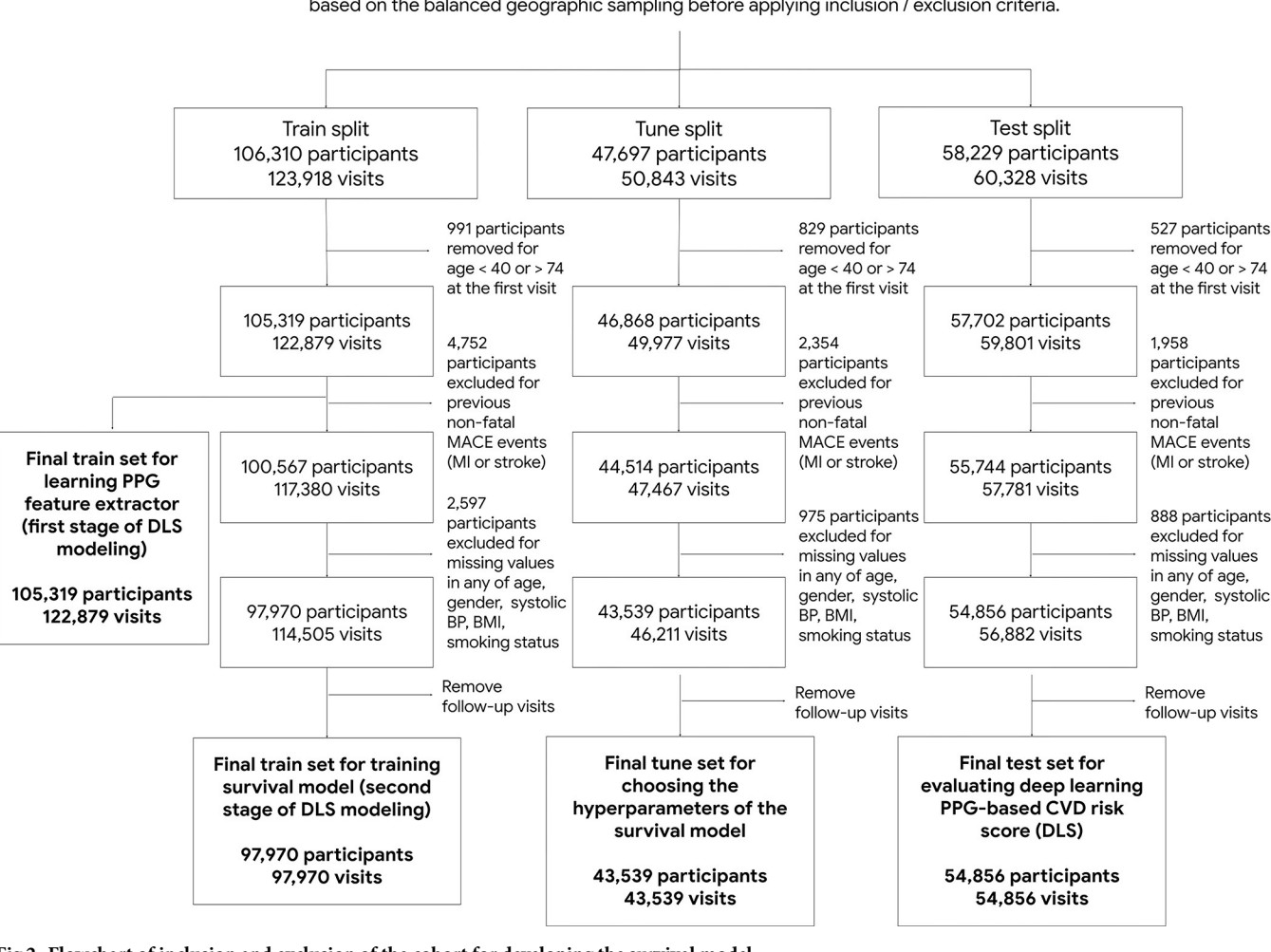

**Fig 2. Flowchart of inclusion and exclusion of the cohort for developing the survival model.**

using a one-dimensional ResNet18 [21, 22, 30] as the neural network architecture. Specifically, the UKB PPG waveform signals (UKB Data-Field 4205) was collected by the PPG device, PulseTrace PCA2 (CareFusion, USA), and the device averaged a minimum of six pulse waveforms with a pulse interval close to the average pulse interval, and output a single summarized waveform stretched to 100 temporal units irrespective of heart rate (for example, see https://biobank.ndph.ox.ac.uk/ukb/refer.cgi?id=100181). Since the UKB normalized the PPG data into 100-length time-series irrespective of the heart rate, the time interval between successive values can vary across participants due to variations in heart rate. Thus, we resampled the x-axis of the raw data so that the interval between two timesteps is uniform (18.2 milliseconds per step), resulting in variable-length time-series per participant. We then padded all samples to the same length with zeros. We also applied the Brownian-tape speed data augmentation (details in the S1 Text, the "Details of model training" section), which is specific to time-series data.

We selected the one-dimensional ResNet18 as the backbone for time-series modeling since one-dimensional convolutional neural networks are known to be a strong model for time-series tasks [31], and one-dimensional ResNet18 is lightweight for potential low-resource setting use, such as deploying on mobile devices.

**Table 1. Cohort statistics for 10-year major adverse cardiovascular events (MACE) risk prediction at the first UK Biobank visit.** Details of the geographic split based on longitude and latitude are listed in S1 Table, S2 Fig.

| | Overall | Train | Tune | Test |
|---|---|---|---|---|
| Number of sites | 15 | 9 | 3 | 3 |
| Geographic location of sites | See train/tune/ test | Swansea, Bristol, Birmingham, Nottingham, Sheffield, Cheadle, Wrexham (Center) | Newcastle, Middlesborough, Liverpool (North-West) | Croydon, Hounslow, Reading (South-East) |
| Patients | 196,365 | 97,970 | 43,539 | 54,856 |
| MACE, count (%) | 5,650 (2.9) | 2,798 (2.9) | 1,401 (3.2) | 1,451 (2.6) |
| Censoring, count (%) | 190,716 (97.1) | 95,173 (97.1) | 42,138 (96.8) | 53,405 (97.3) |
| Age, median [IQR] | 59.0 [51.0,64.0] | 59.0 [52.0,65.0] | 60.0 [52.0,64.0] | 58.0 [50.0,63.0] |
| Sex = female, count (%) | 107,679 (54.8) | 52,885 (54.0) | 23,833 (54.7) | 30,961 (56.4) |
| Smoking status, count (%) | 84,111 (42.8) | 41,559 (42.4) | 18,436 (42.3) | 24,116 (44.0) |
| BMI, median [IQR] | 26.6 [24.0,29.7] | 26.6 [24.1,29.7] | 26.9 [24.3,30.0] | 26.2 [23.7,29.4] |
| SBP, median [IQR] | 136.5 [124.5,149.5] | 137.0 [125.5,150.0] | 139.0 [127.0,152.0] | 133.5 [122.0,146.5] |
| Total cholesterol, median [IQR] | 5.7 [5.0,6.5] | 5.7 [5.0,6.5] | 5.7 [5.0,6.5] | 5.6 [4.9,6.4] |
| HDL, median [IQR] | 1.4 [1.2,1.7] | 1.4 [1.2,1.7] | 1.4 [1.2,1.7] | 1.4 [1.2,1.7] |
| HbA1c, median [IQR] | 35.3 [32.9,38.0] | 35.3 [32.9,37.9] | 35.3 [32.9,37.9] | 35.4 [32.9,38.2] |
| Diabetes, count (%) | 4738 (2.4) | 2129 (2.2) | 1063 (2.4) | 1546 (2.8) |
| Hypertension, count (%) | 52,299 (26.6) | 26,097 (26.6) | 11,618 (26.7) | 14,584 (26.6) |
| History of angina, count (%) | 161 (0.1) | 89 (0.1) | 34 (0.1) | 38 (0.1) |
| History of hyperlipidemia, count (%) | 18,872 (9.6) | 8,517 (8.7) | 4,380 (10.1) | 5,975 (10.9) |
| On hypertension medication, count (%) | 39762 (20.2) | 20072 (20.5) | 9151 (21.0) | 10539 (19.2) |
| On statin, count (%) | 32297 (16.4) | 16068 (16.4) | 7431 (17.1) | 8798 (16.0) |

We trained the feature extractor on the train subset, and picked the network weights that maximized the Cox pseudolikelihood (see description of the second stage below) on the tune subset. These weights were used to compute PPG embeddings on the train, tune, and test subsets. The PPG embeddings were further processed by principal component analysis (PCA)–a technique to reduce dimensionality–to five PCA-derived DLS features that are used by the survival model. Modeling details are listed in S1 Text.

**Second stage: Survival model.** In the second stage, we developed a Cox proportional hazards regression model for predicting ten-year MACE risk, using as inputs age, sex, smoking status, PCA-derived PPG embeddings and PPG-HR (heart rate measured during PPG assessment). The model was trained on the train subset and tuned on the tune subset to decide the best-performing ridge regularization parameter (S2 Table).

**Models for comparison/reference.** For comparisons, we developed different survival models based on different feature sets (Table 2 "Features used" column, and S3 Table), including office-based and laboratory-based refit-WHO scores using the CVD risk predictors adopted in WHO/ISH risk chart and Globorisk studies, metadata-only model (age, sex, smoking status), metadata + PPG morphology (a model with metadata and engineered PPG features describing waveform morphology, such as dicrotic notch presence, details in S4 Table), a model without smoking status as an input (metadata without smoking, DLS without smoking), and the "Full" model that considered metadata, laboratory data, medication and medical history as a reference, to compute CVD risk score (details in the S2 Text). We chose the model using shared predictors from the office-based WHO/ISH risk chart and Globorisk score (office-based refit-WHO score) as the main reference since they are adopted in the CVD risk

**Table 2. Model performance comparison of 10-year major adverse cardiovascular events (MACE) risk prediction between DLS versus other methods for the non-operating point dependent metrics.**

| Model | C-statistic (%) | Delta (%) | P-value for non-inferiority of C-statistic | P-value for superiority of C-statistic | cfNRI (%) | cfNRI (event) (%) | cfNRI (non-event) (%) | Calibration slope | Features used* |
|---|---|---|---|---|---|---|---|---|---|
| Office-based refit-WHO | 70.9 (69.7, 72.2) | | n/a (reference) | | | | | 0.979 (0.915, 1.038) | Metadata + BMI + SBP |
| DLS | 71.1 (69.9, 72.4) | 0.2 (-0.4, 0.8) | <0.01 | 0.292 | 0.1 (0.0, 0.1) | 0.1 (0.0, 0.2) | 0.0 (0.0, 0.0) | 0.981 (0.919, 1.045) | Metadata + PPG |
| Metadata | 69.1 (67.9, 70.4) | -1.7 (-2.2, -1.3) | <0.01 | 1 | -0.4 (-0.5, -0.3) | -0.2 (-0.3, -0.2) | 0.2 (0.1, 0.2) | 0.94 (0.875, 1.004) | Metadata |
| SBP-140 | 59.4 (58.3, 60.5) | -11.5 (-12.7, -10.2) | 1 | 1 | -1.3 (-1.4, -1.2) | -1.0 (-1.1, -0.9) | 0.3 (0.3, 0.3) | - | SBP |
| Evaluated on the subset with all PPG morphology data available | | | | | | | | | |
| Metadata + PPG morphology | 70.0 (68.8, 71.3) | -0.9 (-1.4, -0.4) | <0.01 | 1 | -0.1 (-0.2, -0.1) | -0.1 (-0.1, 0.0) | 0.1 (0.1, 0.1) | 1.02 (0.951, 1.086) | Metadata + engineered PPG features |
| Office-based refit-WHO | 70.9 (69.7, 72.2) | | n/a (subset reference) | | | | | 0.977 (0.913, 1.035) | (Metadata + BMI + SBP) |
| Evaluated on the subset with laboratory data available | | | | | | | | | |
| Lab-based refit-WHO | 71.6 (70.4, 72.9) | 0.5 (0.1, 0.9) | <0.01 | <0.01 | 0.2 (0.1, 0.2) | 0.3 (0.2, 0.3) | 0.1 (0.1, 0.1) | 0.921 (0.864, 0.982) | Metadata + total cholesterol + glucose |
| Office-based refit-WHO | 71.1 (69.9, 72.4) | | n/a (subset reference) | | | | | 0.897 (0.838, 0.959) | (Metadata + BMI + SBP) |

The primary analysis of the study is non-inferiority of the C-statistic of the DLS model compared with the office-based refit-WHO model. 95% confidence intervals (CIs) of C-statistic, cfNRI, and slope were obtained via bootstrapping, and the p-values were computed via a permutation test. The slope was not calculated for SBP-140 since its output is binary. One out of five PPG features in the DLS model yielded a p-value < 0.05 for the non-proportional test, though this is not statistically significant after correcting for multiple testing (alpha of 0.01). As a sensitivity analysis, removing or binning this PPG feature yielded a C-statistic of 72.0 (70.8, 73.2) and 71.9 (70.8, 73.2), respectively. *In the "Feature used" column, "Metadata" includes age, sex, and smoking status.

research for low-resource settings. To ensure the fairest comparison the coefficients for the WHO and Globorisk predictors were re-fitted using the same UKB train subset as our DLS, and a sensitivity analysis was conducted using the original coefficients with recalibration.

We further developed DLS+ (DLS with BMI), and DLS++ (DLS with BMI and SBP) that additionally included more non-laboratory, office-based measurements as inputs of the survival model to better understand the prognostic value of PPG on top of the existing office-based refit-WHO model.

All models were trained on the same train subset and tuned on the tune subset except for laboratory-based refit-WHO score, metadata + PPG morphology, and the Full models that we trained, tuned and compared based on a subset of the testing data without missing values of the input features.

## Evaluation

**Endpoints.** The outcome, ten-year risk of MACE, was defined as a composite outcome of three components, non-fatal myocardial infarction, stroke, and CVD-related death (using ICD codes and cause of death to identify, S4 Table for details) [7, 32]. To define the outcome, we used (1) the date of heart attack, myocardial infarction, stroke, ischemic stroke, either diagnosed by doctor or self-reported, (2) the record of ICD-10 (international classification of diseases, 10th revision) clinical codes, and (3) and strings that are associated with the CVD-

related death. The ICD-10 codes used included I21 (acute myocardial infarction), I22 (subsequent myocardial infarction), I23 (complications after myocardial infarction), I63 (cerebral infarction), I64 (stroke not specified as hemorrhage or infarction). The strings we used for matching include those related to coronary artery diseases, myocardial infarction, stroke, hypertensive diseases, heart failure, thromboembolism, arrhythmia, valvular diseases and other heart problems. We used the earliest date on any of the data sources mentioned above as the outcome date.

**Statistical analysis.** For primary analysis, we compared DLS with the office-based refit-WHO score, which is a risk model for healthy individuals across different countries [4, 7, 33, 34], using Harrell's C-statistic. We conducted a non-inferiority test with a pre-specified margin of 2.5% and alpha of 0.05, both selected based on power simulations using the tune subset. For secondary analyses, we also compared DLS with scores generated by other models mentioned in "Models for Comparison/Reference" above.

Additional evaluation metrics included the category-free net reclassification improvement (cfNRI) [35], and after defining a specific risk threshold (model operating point), sensitivity, specificity, NRI, and adjusted hazard ratio (HRs). For NRI and cfNRI, we also reported the respective event and non-event components. Risk thresholds were selected in three ways: (1) matching the sensitivity of SBP-140 (described next), (2) matching the specificity of SBP-140, and (3) the 10% predicted risk threshold suggested by the Globorisk study [4]. Elevated SBP above 140 mmHg ("SBP-140") [36] was used for threshold selection because it is used as a simple single-visit indicator of BP control in the healthcare program of some countries such as India [37], and we hypothesized that the PPG provided a single-visit indicator of vascular properties. To calculate sensitivity and specificity, we excluded the participants without a ten-year follow up if they didn't have a MACE event within ten years. To evaluate model calibration, we used the slope of the line comparing predicted and actual event rates, for deciles of model prediction [33]. We also performed subgroup analyses based on smoking status, sex, age, elevated HbA1c and hypertension status. We used quintiles for the elevated HbA1c subgroup due to the smaller sample size.

For statistical precision, we used the Clopper-Pearson exact method to compute the 95% confidence intervals (CIs) for sensitivity and specificity, and used the non-parametric bootstrap method with 1,000 iterations to compute 95% of all remaining metrics and delta values. For hypothesis tests in secondary and exploratory analysis, we used a permutation test to examine the non-inferiority and superiority of the C-statistic, and the one-sided Wald test for sensitivity and specificity. The log-rank test was used to determine whether survival differs between the model-defined low and high risk groups. For all two-sided tests, we used an alpha value of 0.05.

The deep learning framework (JAX) used in this study is available at https://github.com/google/jax [38]. All survival analyses were conducted using Lifelines [39], an open source Python library.

## Ethics statement

This study involving retrospective de-identified data was reviewed by and granted a waiver by the Advarra Institutional Review Board. All participants in the UK Biobank study gave broad consent to use their anonymized data and samples for any health-related research [40].

## Results

We showed that DLS demonstrated non-inferiority to the office-based refit-WHO score. We evaluated the ten-year MACE risk prediction performance of all methods using our UKB test

subset, which was held-out during the training process. The DLS yielded C-statistic of 71.1% (95% CI [69.9, 72.4]). When compared with the office-based refit-WHO score, the DLS was non-inferior (p<0.01), with a delta of +0.2% (-0.4, 0.8). The cfNRI was 0.1% (0.0, 0.1), stemming primarily from improved reclassification of events (0.1% [0.0, 0.2]), without performance penalty in the non-events (0.0% [0.0, 0.0]).

Based on the C-statistic, there was an incremental improvement when the metadata model (69.1%) was augmented with manually engineered (not deep learning derived) PPG morphology features (70.0%). The DLS was superior to this metadata+PPG morphology features model (p<0.01), indicating value in deep learning based feature extraction. The lab-based model (which requires total cholesterol and glucose information) was superior to the office-based refit-WHO score (71.6% versus 70.9%, p<0.01). With only five deep learning-based PPG features, the C-statistic is 62.7% (61.3, 63.9). By applying the Globorisk scores in [7], which recalibrating on UKB cohort for baseline hazard and mean risk factors but without re-estimating the coefficients, the office-based Globorisk yielded a C-statistic of 70.0% (68.8, 71.2), and the lab-based Globorisk yielded a C-statistic of 69.8% (68.5, 71.1). More details are shown in Table 2.

For a fair comparison, we then selected the risk thresholds that matched the specificity or sensitivity of SBP-140 (see Statistical analysis in Methods) (specificity of 63.7%, sensitivity of 55.2%). We found that at matched specificity, the sensitivity of the DLS (67.9%) were non-inferior to the office-based refit-WHO score (67.7%) (p = 0.012), and a comparable NRI, while the metadata and metadata + PPG morphology models were not (p = 0.984 and p = 0.305, respectively). At the matched sensitivity, the DLS's specificity (74.0%) was also non-inferior to the baseline (73.1%) (showed superiority with p<0.01), with a comparable NRI. The laboratory-based refit-WHO and the model using metadata and PPG morphology-based features were also non-inferior to the office-based refit-WHO score, despite these models requiring additional inputs from laboratory measurements or engineered PPG features, respectively. The metadata-only model performed more poorly than the office-based refit-WHO score across different metrics. More details are shown in Table 3. We also conducted Kaplan Meier analysis on risk groups defined using the above approach (Fig 3). Both thresholds showed significant (p<0.01, log rank tests) differences between the groups. Results for the 10% risk threshold are in S5 Table and S3 Fig.

In addition to the default set of inputs to the DLS, we also evaluated models with BMI, and with both BMI and SBP (both of which are predictors in the office-based refit-WHO) included as additional inputs, which we refer to as DLS+ and DLS++, respectively. We found that adding BMI (DLS+) improved DLS in terms of both discrimination and net reclassification. Additional improvement was observed after adding SBP (DLS++), which further improved the DLS model, and demonstrated superiority across different metrics (S6 Table). We also showed that for DLS and its variants (DLS+ and DLS++), the cfNRI and NRI with different risk thresholds (S6A Table) were also on par with the office-based refit-WHO score (S6B Table). These findings indicate that combining the existing non-laboratory risk factors from the refit-WHO score with the DLS features yields a more accurate CV risk estimation. We further developed a model (Full model) that includes more risk factors used in the CVD risk scores commonly used in high-income countries (QRISK and/or ASCVD), as well as a model that incorporates genetic risk, and listed the findings in the S2 Text.

Meanwhile, the predicted and observed risks of ten-year MACE were similar across different models (Fig 4), which indicated DLS has similar calibration performance compared with other models. The calibration slope of DLS was similar to the office-based refit-WHO score (0.981 versus 0.979) (Table 2). We also found that DLS++ has a comparable calibration

**Table 3. Model performance comparison of 10-year major adverse cardiovascular events (MACE) risk prediction between DLS versus each of other methods using a risk threshold matches the same specificity or sensitivity of SBP-140.**

| Model | Sensitivity@specificity of 63.7% | | | | | | | Specificity@sensitivity of 55.2% | | | | | | |
|---|---|---|---|---|---|---|---|---|---|---|---|---|---|---|
| | Mean (%) | Delta (%) | Non-inferiority p-value | Superiority p-value | NRI (%) | NRI (event) (%) | NRI (non-event) (%) | Mean (%) | Delta (%) | Non-inferiority p-value | Superiority p-value | NRI (%) | NRI (event) (%) | NRI (non-event) (%) |
| Office-based refit-WHO | 67.7 (65.2, 70.1) | reference | | | | | | 73.1 (72.7, 73.5) | reference | | | | | |
| DLS | 67.9 (65.4, 70.3) | 0.1 (-1.9, 2.0) | <0.01 | 0.654 | -0.3 (-2.0, 1.6) | 1.0 (-0.9, 2.9) | 1.2 (0.9, 1.5) | 74.0 (73.6, 74.4) | 0.9 (-0.7, 2.5) | <0.01 | <0.01 | 1.1 (-0.9, 3.1) | 1.6 (-0.5, 3.4) | 0.4 (0.1, 0.8) |
| Metadata | 63.6 (61.0, 66.0) | -4.1 (-5.9, -2.2) | 0.984 | 1 | -2.7 (-4.5, -0.9) | 0.8 (-0.9, 2.6) | 3.5 (2.9, 3.8) | 70.4 (70.0, 70.8) | -2.4 (-3.9, -1.1) | 0.961 | 1 | -2.4 (-4.9, -0.3) | -0.4 (-3.1, 1.6) | 2.0 (0.6, 2.7) |
| SBP-140 | 55.4 (53.1, 57.9) | -12.4 (-15.1, -9.3) | 1 | 1 | -11.1 (-13.6, -8.2) | -18.7 (-21.2, -15.8) | -7.7 (-8.1, -7.2) | 63.6 (63.2, 64.0) | -11.0 (-74.4, -8.5) | 1 | 1 | -6.8 (-31.6, -3.1) | 12.6 (8.9, 27.8) | 16.6 (16.0, 56.5) |
| *Evaluated on the subset with all PPG morphology data available* | | | | | | | | | | | | | | |
| Metadata + PPG morphology | 66.0 (63.5, 68.5) | -1.8 (-3.5, -0.1) | 0.305 | 0.992 | -1.6 (-3.4, 0.2) | 0.3 (-1.5, 2.2) | 1.9 (1.7, 2.2) | 71.7 (71.3, 72.0) | -1.5 (-2.9, -0.1) | <0.01 | 1 | -1.5 (-3.7, 0.7) | -0.2 (-2.4, 2.0) | 1.3 (1.1, 1.6) |
| Office-based refit-WHO | 67.7 (65.3, 70.1) | n/a (subset reference) | | | | | | 73.1 (72.8, 73.5) | n/a (subset reference) | | | | | |
| *Evaluated on the subset with laboratory data available* | | | | | | | | | | | | | | |
| Lab-based refit-WHO | 69.1 (66.6, 71.6) | 1.0 (-0.7, 2.5) | <0.01 | 0.106 | 2.0 (0.3, 3.7) | 4.2 (2.5, 5.8) | 2.2 (1.9, 2.4) | 74.8 (74.4, 75.2) | 1.5 (0.1, 2.5) | <0.01 | <0.01 | 2.7 (1.1, 4.4) | 4.3 (2.6, 6.0) | 1.5 (1.3, 1.8) |
| Office-based refit-WHO | 68.2 (65.7, 70.7) | n/a (subset reference) | | | | | | 73.4 (73.0, 73.8) | n/a (subset reference) | | | | | |

95% confidence intervals (CIs) of sensitivity and specificity were obtained from the Clopper-Pearson exact method, and the p-values were calculated by the permutation test with the prespecified margin of 2.5% and alpha of 0.05. The 95% CIs of NRI were computed via bootstrapping.

performance (S6 Table). All models except the DLS+ have an observed ten-year MACE risk estimation within 5% mean absolute calibration error (i.e., the slopes were between 0.95–1.05).

Finally, DLS is on par with the office-based refit-WHO score in some subgroups. S7 Table shows that DLS demonstrated non-inferiority in some subgroups and showed superiority in smoking, hypertensive and male subgroups. Both the office-based refit-WHO score and DLS had similar performance trends. Both models have higher sensitivity and lower calibration error but lower specificity on the smoking, older, male, and hypertensive subgroups. The models were well-calibrated for most subgroups, but systematically overestimated absolute risk about 4.0% in the elevated A1c and about 1.0% in hypertensive subgroups. The finding indicates that the developed risk models tend to be better calibrated and better predict ten-year MACE risk in a population that has higher known CVD risk factors, such as older, male, smoking, higher blood glucose and hypertensive subgroups (S7 Table). Across different age, sex, smoking, and comorbidity (diabetes and hypertension) subgroups, the calibration for all risk scores were similar in predicting ten-year MACE risk in smoking, age greater than 55, male, not elevated A1c populations, with prediction errors within 10% (i.e. the calibration regression slope between 0.9 and 1.1 (S7 Table, S4 Fig)).

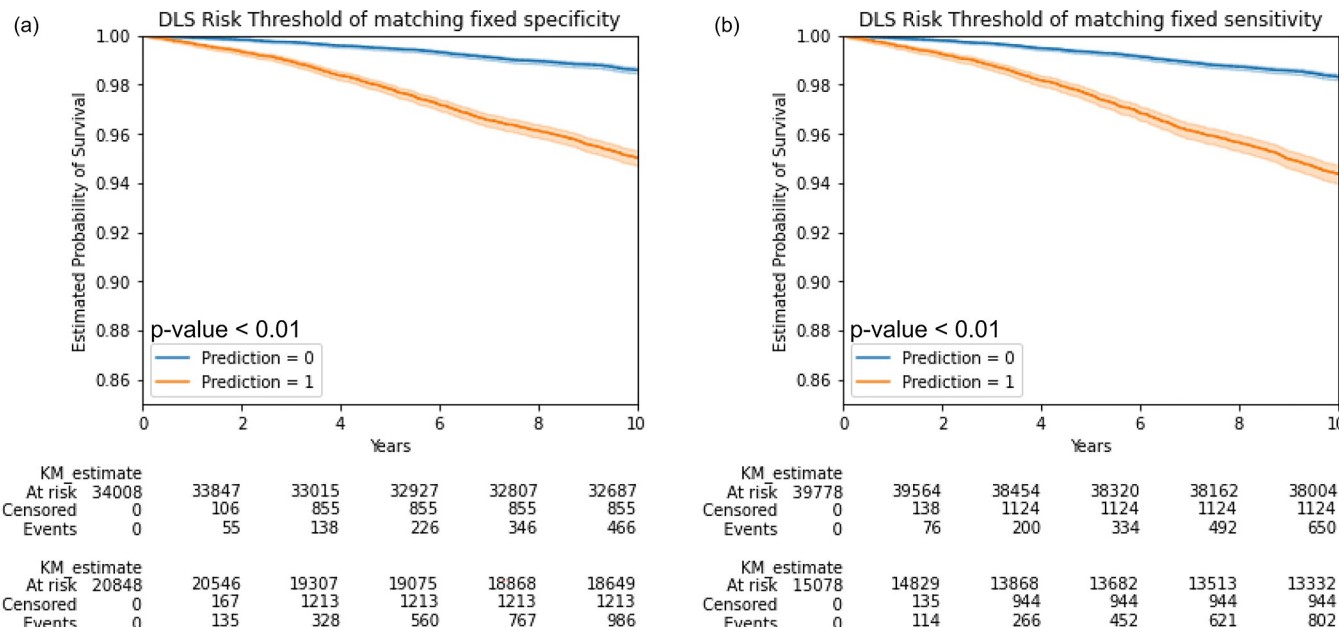

**Fig 3. Kaplan-Meier curves for the DLS with different definitions of high risk.** (A) Risk threshold corresponding to a specificity of 63.6%, (B) Risk threshold corresponding to a sensitivity of 55.4% (see Methods). The p-values were calculated by the log-rank test.

To understand the DLS model deeper, we analyzed the DLS-based PPG features via coefficients and hazard ratios (HRs), correlations with known PPG morphological features, the difference of PPG waveforms between high/low values of DLS PPG feature, and saliency map of the PPG features, computed using integrated gradients [41] of the Cox log partial hazard with respect to the input waveform, using the same metadata and linear interpolation from a constant (all zeros) baseline for the PPG. We first examined the association between each model and MACE via the coefficients and hazard ratios (HRs) (S8 Table). We found that in the office-based refit-WHO score, smoking, older age, higher BMI, and higher SBP were associated with ten-year MACE risk. We found that some DLS features were also associated with ten-year MACE risk ($p<0.05$ for four deep learning PPG features in DLS and DLS+, and for two PPG features in DLS++). Next, we computed the Spearman's rank correlation coefficient between DLS features and engineered PPG morphological features (Table 4), and visualized the relationship between waveforms and the PPG feature values / predicted risk score change, along with the integrated gradients.

In Fig 5, we have visualized the PPG waveforms based on the predicted risk score and five DLS PPG feature values, by presenting the average of 100 PPG waveforms sampled nearest to the following quantiles: 10th, 50th, and 90th. We found that with higher PPG-1 value, the systolic peak shifts earlier; notch appears more prominent. A leftward systolic peak shift is consistent with a higher slope and thus less stiff vessels, corroborated by the strong positive correlation with the peak-to-peak time feature and negative correlation with the stiffness index. Together, these findings are consistent with the risk prediction observations above because PPG-1 has the highest absolute Cox coefficient among the five DLS PPG features, and the direction is negative (hence inverse direction with the observations in the predicted score). PPG-2 is also correlated with peak-to-peak time and stiffness index, and the cases with higher PPG-2 values show a lower waveform and notch amplitude, which may be consistent with the negative correlation with reflection index, though other morphological changes along this spectrum are harder to describe precisely. PPG-3 is weakly correlated with cardiac events (it

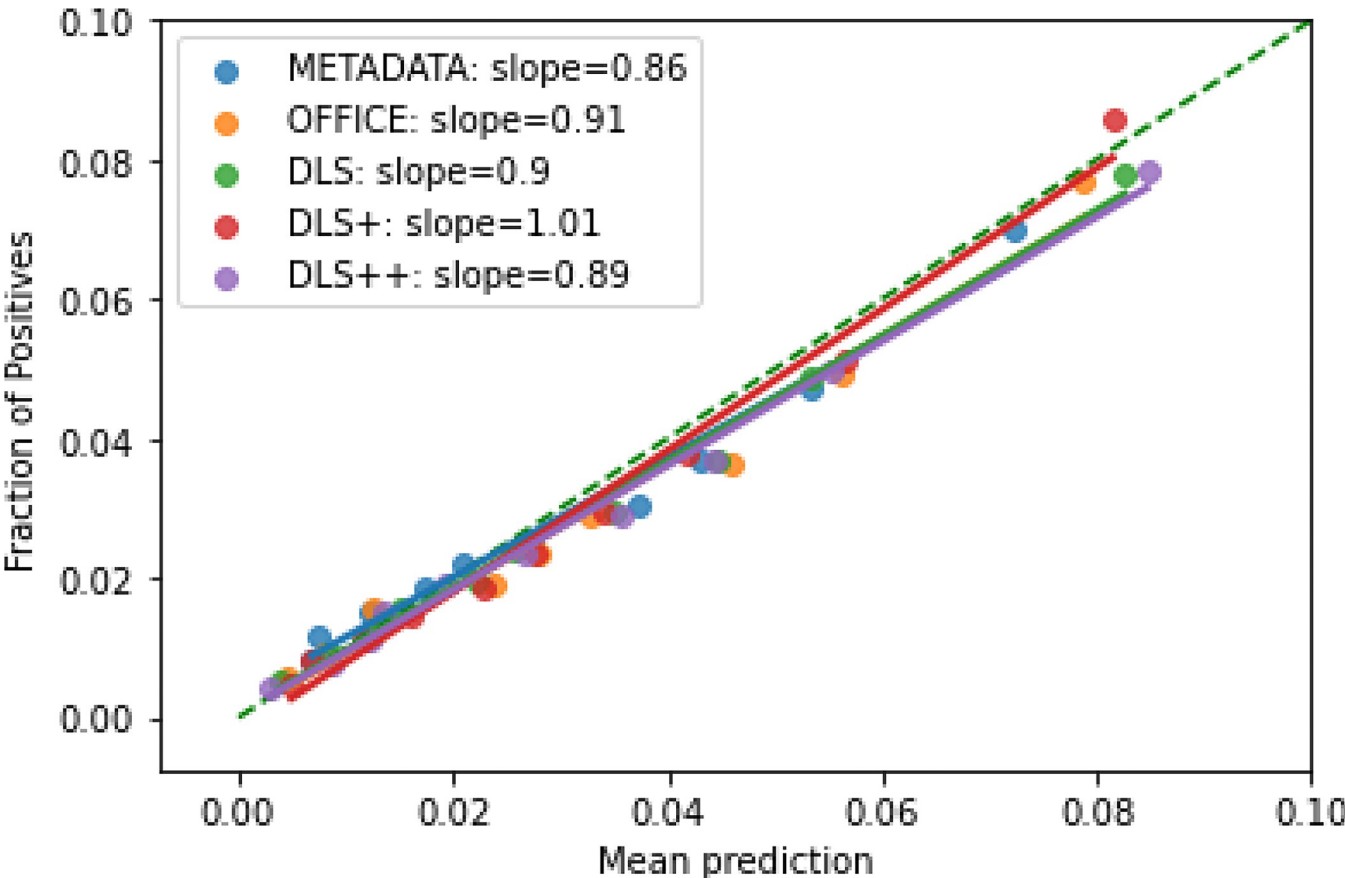

**Fig 4. Calibration plot, showing observed and predicted 10-year MACE risk.** We discretized each model's output into deciles and the slopes indicate the coefficient of a linear regression. METADATA: the risk model with age, sex, smoking status), OFFICE: the risk model using shared predictors from the office-based WHO/ISH risk chart and Globorisk score. DLS: the risk model with metadata and deep learning-based PPG features. DLS+: the risk model using all DLS predictors plus BMI. DLS++: the risk model using all DLS predictors plus BMI and systolic blood pressure.

has a small Cox coefficient), and weakly correlated with the available UK Biobank-provided PPG features, we also didn't see a significant difference between waveforms with the change of PPG-3 value. PPG-4 is correlated with the shoulder position and more weakly, with systolic peak position. The morphological differences at the notch are subtle, though the leftward shift is consistent from 10th to 50th to 90th percentile. PPG-5 appears correlated to changes in the relative notch height, which is consistent with its high correlation with reflection index. The reflection index is in turn a measure of peripheral resistance, and it also tends to be more common in some specific age group (40–54) and obese groups [42], and it also loosely correlates with other risk predictor metric such as pulse wave velocity (PWV) [43]. Interestingly, PPG-5 is the only PPG feature which had a positive univariable association (positive Cox coefficient) with MACE outcomes, but which became a negative association in multivariable analysis. This likely reflects its effect being that of modulating other features. We also found that stratifying by the DLS predicted risk score, the systolic peak shifts later with higher risk prediction and notch appears to be less prominent, which is also related to vascular stiffness. Finally, we found that saliency maps, computed using integrated gradients, (Fig 5 right) highlighted the waveform's peak, notch and diastolic phase as the areas most responsible for changes in predicted risk.

**Table 4. Spearman's rank correlation coefficients between each PPG feature and annotated PPG features from the UK Biobank.**

| DLS PPG Feature | Univariable Cox coefficient for DLS feature | Multivariable Cox coefficient for DLS feature | Correlation of DLS feature with UK Biobank PPG feature | | | | | | | Comments |
|---|---|---|---|---|---|---|---|---|---|---|
| | | | Pulse rate (0.077)* | Reflection index (0.052)* | Peak-to-peak time (-0.232)* | Position of the peak (0.281)* | Position of the notch (-0.009)* | Position of the shoulder (0.281)* | Stiffness index (0.027)* | |
| PPG-1 | -0.154 | -0.111 | 0.074 | -0.16 | 0.48 | -0.48 | 0.061 | -0.539 | -0.43 | Correlation with peak position, systolic peak width, narrow clearer notch |
| PPG-2 | 0.034 | 0.059 | 0.18 | -0.438 | 0.373 | -0.002 | 0.293 | -0.177 | -0.418 | Correlation with peak-to-peak time |
| PPG-3 | 0.065 | 0.002 | 0.114 | -0.114 | 0.068 | -0.127 | 0.121 | -0.021 | -0.033 | Weakly correlated to known PPG features |
| PPG-4 | 0.010 | 0.032 | 0.147 | -0.121 | -0.095 | 0.273 | 0.28 | 0.354 | 0.066 | Correlation with shoulder and peak position |
| PPG-5 | 0.063 | -0.055 | -0.041 | 0.227 | 0.02 | -0.047 | 0.012 | 0.024 | 0.032 | Correlation with reflection index |

The comments column summarizes observations about these features. Pulse rate is the heart rate during arterial stiffness measurement. Reflection index is a measure of reflection, the ratio between reflected wave peak and the direct wave peak height. Peak-to-peak time is the difference between the peak values of direct and reflected components. Peak position is the time when the peak occurs in the waveform. Notch position is the time when notch occurs in the waveform. Shoulder position is the time of the shoulder-like curve in the waveform, and Stiffness index is calculated by the peak-to-peak time divided into the person's height. Details of UKB features are listed in S4 Table.

Univariable Cox coefficients of the UK Biobank PPG feature after normalizing to zero mean and unit variance, to help understand directionality of association with MACE.

## Discussion

We developed a deep learning PPG-based CVD risk score, DLS, to predict ten-year MACE risk using age, sex, smoking status, heart rate and deep learning-derived PPG features. Without requiring any vital signs or laboratory measurement, DLS demonstrated non-inferior performance compared to the office-based refit-WHO score with coefficients re-estimated on the same cohort. Results were consistent between metrics (C-statistic, NRI, cfNRI, sensitivity, specificity, calibration slope), and in various subgroups. Improved cfNRI and NRI also indicate the capability of DLS to reclassify cases better than the office-based refit-WHO score. Additionally, if available, adding office-based features (BMI, SBP) on top of DLS further improved the model performance.

Our work focuses on understanding the role that PPG and deep learning can play in settings where equipment access to healthcare is limited, such as community-based screening programs in LMICs. Several CVD prediction scores without an assumption of the availability of laboratory measurement exist for primary prevention, such as WHO/ISH risk prediction chart [34], office-based Framingham risk score (FRS) [44], office-based Globorisk score [4], non-laboratory INTERHEART risk score [45], and Harvard NHANES risk score [46]. Some of these are also deployed in real-world clinical practice [4, 32], though these methods require either body measurements (BMI, waist-hip ratio), SBP, or both. Challenges remain in scaling up CVD screening in the resource-limited areas due to reasons such as the lack of laboratory devices, sphygmomanometer cuffs, or the necessary training of CHWs for accurate measurements. In our study, the DLS demonstrated performance comparable to that of the re-estimated office-based refit-WHO score, without requiring accurate laboratory examination, vital

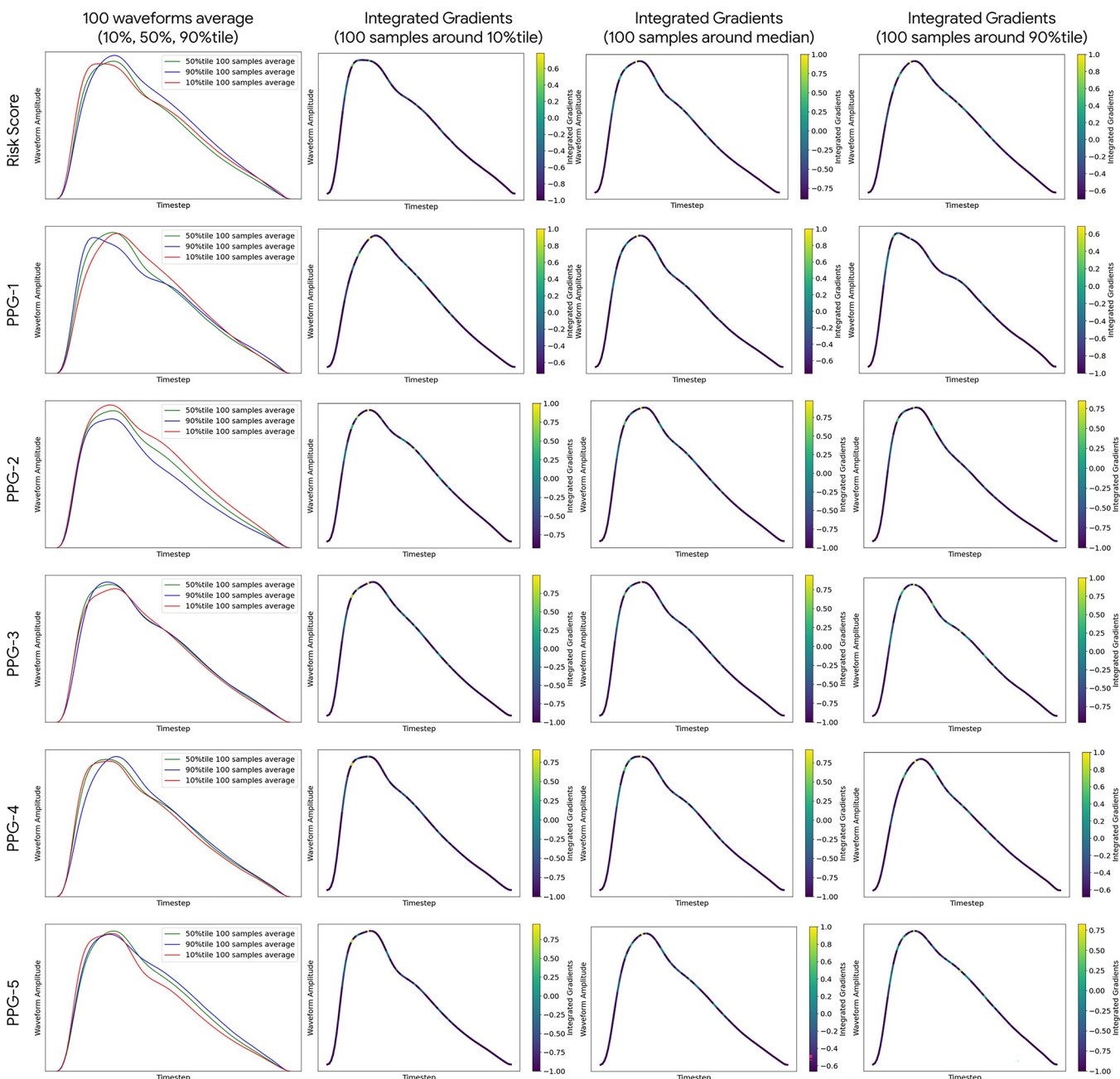

**Fig 5. DLS PPG features explainability visualizations.** The first row sorts PPGs by predicted risk, whereas the next 5 rows sort PPGs based on the 5 PPG feature values. The first column presents the average of 100 PPG waveforms sampled nearest to the following quantiles of the quantity mentioned on the left: 10th (red) 50th (green) and 90th (blue). The next 3 columns present the respective averaged PPGs along with the normalized saliency values based on integrated gradients. We observed that in general salient areas that most influence the predictions seem to be near the top of the systolic peak and the notch, independent of which quantile and feature/prediction the PPG was sampled from. Each PPG feature appears to correspond to different morphological aspects (see Table 4).

signs measured via additional devices, or BMI. This feature improves accessibility for health systems that have limited resources to collect vitals and labs for CVD risk screening and triage. More intriguing, PPG signals could in principle be captured through a smartphone [16], and future work could leverage smartphone-based PPGs along with the DLS to enable large-scale screening and triage in the community at low cost (Fig 1) [14, 47].

Due to the higher prevalence, lower diagnosis rate and lower treatment of CVD in LMICs, WHO has listed preventing and controlling CVD as main targets in their "Global action plan for the prevention and control of non-communicable diseases (NCDs) 2013–2030" [48]. PPG-based screening may allow healthcare systems to optimize use of resources by funneling in those who are likely to benefit the most and improve the early detection of CVDs. Thus, our study represents a step on the journey towards enabling community-based preventive treatment for high CVD risk individuals with limited healthcare access.

The deep learning-based features are challenging to interpret directly, and the pathophysiology between PPG and CVD risk is still under investigation [49]. Our analysis, including the hazard ratios, correlations, feature values, and saliency maps, provides interpretability insights, suggesting the DLS-extracted features reflect morphologic changes in the waveform independent of heart rate. However, further investigation is still required for interpreting the deep learning-based features and whether experts can learn from these features. We also found preliminary evidence that using a resting electrocardiogram (ECG) yielded a comparable performance to the office-based model on a UKB subset with available resting ECG data.

Several limitations of the study should also be noted. We used a single dataset, UKB, for both modeling and evaluation. Though we have stratified the UKB cohort based on geographical information to allow for non-random variation [29], further work is needed to understand generalization to other populations. Notably, UKB is not representative of the population in LMICs. However, using UKB to demonstrate the capability of using DLS for long-term CVD risk prediction is an important first step in justifying a prospective data collection in LMICs. The device used for PPG acquisition across the UKB is a specific clinical pulse oximeter (PulseTrace PCA2), thus our results provide direct evidence that waveforms from this pulse oximeter may be a reliable CVD screening tool. Studies have found that the heart rate and rhythm extracted from smartphone PPG were comparable with clinical grade devices such as ECG [50–52], but additional work is needed to know if our model would transfer to smartphone-collected PPGs. Since the PPG waveform signals in the UKB have only been collected by a single device and specific protocol (details in Model development section), further work may be necessary to understand if these data are biased in some way (e.g., less noisy) relative to less structured data collection protocols. However, our results on UKB indicate that using PPG for CVD screening and triaging is promising and worth investigating further, particularly in lower-resourced regions. Because a dataset of PPG collected using commodity devices such as smartphones does not yet exist, to help with this, we are open-sourcing a sample PPG data collection app (see Data Availability Statement) to facilitate the future relevant data collection and similar longitudinal PPG research studies in LMICs. Future work may be needed to understand how to mitigate any differences in PPG features based on PPG device or manufacturer. We are also releasing the trained PPG embeddings via the UK Biobank, and the analysis code via the GitHub repository (see Data Availability Statement). Future work could focus on predicting CVD risk using prospective smartphone PPG datasets from low-resource healthcare systems. Additional work will also be needed to know if our model would transfer the smartphone setup, because the smartphone PPG datasets with longitudinal MACE outcomes in LMIC do not exist to our knowledge, so direct evidence of the efficacy of such an approach will need to be evaluated when such data become available. We have also investigated other larger network architectures such as the dilated CNN and WaveNet yet without marked performance improvement, thus further investigation on the efficient modeling that can improve the performance yet also being lightweight will be considered.

To summarize, our study found that a deep learning model extracted features that when added to easily extractable clinical and demographic variables (such as smoking status, age and sex), provided statistically significant prognostic information about cardiovascular risk. Our

work is an initial step towards accurate and scalable CVD screening in resource-limited areas around the world, and will hopefully inspire the collection of real-world datasets with smartphone-acquired PPG and longitudinal outcomes.

## Supporting information

**S1 Fig. Overview of our deep learning-based risk prediction model, DLS.** Blue: models; yellow: inputs; red: intermediate data representations (embeddings) obtained from the deep learning-based PPG feature extractor.
(DOCX)

**S2 Fig. Geographical location information of sites visualized by longitude and latitude for dataset splits.**
(DOCX)

**S3 Fig. Kaplan-Meier estimation of DLS with different operating points.** We compared the survival estimation between the high and low risk groups, which were defined by the risk threshold at 10% suggested by the Globorisk study [1]. For example, a case with prediction value higher than 0.1 will be high risk, else low risk. The p-values were calculated by the log-rank test.
(DOCX)

**S4 Fig. Calibration plots for all subgroups.** The calibration slope values indicate the coefficient of a linear regression where the dependent variable was the fraction of positives (predicted risk) and the independent variable was the mean prediction. We used ten bins to discretize the prediction interval and chose deciles of predicted risk to define the widths of the bins. For the elevated HbA1c subgroup, we used quintiles to ensure sufficient events. All models (office-based refit-WHO, DLS) are calibrated better in smoking, older, male, non A1c elevated, and non-hypertensive subgroups.
(DOCX)

**S5 Fig. Prevalence of major adverse cardiovascular event (MACE) in individuals according to model-predicted risk percentiles.** For each of four risk models, the prevalence of MACE was computed in the individuals scoring in the highest 20, 10, and 5% risk according to the model. Error bars computed via 100 bootstrap iterations. The dashed gray line shows MACE prevalence in the entire sample. Metadata+, model containing age, sex, smoking status, and BMI. DLS+, model containing age, sex, smoking status, BMI, and PPG. Metadata+ + polygenic risk score (PRS), model containing age, sex, smoking status, BMI, and polygenic risk score. DLS+ + PRS, model containing age, sex, smoking status, BMI, PPG, and PRS.
(DOCX)

**S1 Text. Supporting methods.**
(DOCX)

**S2 Text. Supporting results.**
(DOCX)

**S1 Table. Geographical location information of sites for split division.**
(DOCX)

**S2 Table. Training setup for the photoplethysmography (PPG) feature extractor.**
(DOCX)

**S3 Table. Features used in different models for comparison.** We compared all methods with the office-based refit-WHO model. The evaluations of DLS models and additional reference methods are in the main content, S5, S6 and S10 Tables. For the supporting reference methods, the results are listed in the Supporting Tables. *Lab-based refit-WHO and metadata + PPG morphology models are compared with a subset of the whole cohort. **The full model used most QRISK features. The detail of the feature set is described in the Supporting Methods.
(DOCX)

**S4 Table. UK Biobank variables used in the study.**
(DOCX)

**S5 Table. Model performance comparison of 10-year major adverse cardiovascular event (MACE) risk prediction between DLS versus other methods at the 10% risk threshold.** The sensitivity, specificity, and net reclassification improvement (NRI) were calculated at the 10% risk threshold suggested by the Globorisk study for the British population [1]. CIs of sensitivity and specificity were obtained from the Clopper-Pearson exact method, and the p-values were calculated by the permutation test with a prespecified margin of 2.5% and alpha of 0.05. The 95% CIs of NRI were computed by bootstrapping.
(DOCX)

**S6 Table. Model performance comparison of 10-year major adverse cardiovascular events (MACE) risk prediction between DLS versus DLS+ (adding BMI) and DLS++ (adding BMI and SBP).** (a) We examined the discrimination performance using C-statistic, reclassification improvement using category-free net reclassification improvement (cfNRI), and model calibration using the slope value from the reliability diagram. *In "Feature used" column, "Metadata" includes age, sex, and smoking status. (b) The sensitivity was calculated at the risk threshold matching the specificity of SBP-140, and the specificity was calculated at the risk threshold matching the sensitivity of SBP-140. 95% confidence intervals (CIs) of C-statistic, cfNRI, and slope were obtained from the bootstrapping, and p-values were computed by the permutation test. CIs of sensitivity and specificity were obtained from the Clopper-Pearson exact method, and the p-values were calculated by a permutation test with the prespecified margin of 2.5% and alpha of 0.05. The 95% CIs of NRI were computed by bootstrapping.
(DOCX)

**S7 Table. Comparison of 10-year major adverse cardiovascular event (MACE) risk prediction performance between different subgroups using DLS versus office-based refit-WHO model.** The sensitivity and specificity were calculated at the risk threshold matching SBP-140's specificity (see Statistical Analysis). 95% confidence intervals (CIs) were obtained from the Clopper-Pearson exact method.
(DOCX)

**S8 Table. Coefficients and hazard ratios from the Cox's models for 10-year major adverse cardiovascular events (MACE) risk prediction on the UK Biobank (UKB) cohort using DLS, DLS+ and DLS++.** Hazard ratios are shown at the median age of the MACE event, which is 63 years in the train split of UKB cohort. Hazard ratios for smokers are for men, and their interaction with sex shows the adjusted risk for women. We included interaction terms between age and other predictors because the HRs for proportional effects on CVD declined with age [2, 3].
(DOCX)

**S9 Table. The list of proxy tasks used for multitask learning.**
(DOCX)

**S10 Table. Model performance comparison of 10-year major adverse cardiovascular event (MACE) risk prediction between the office-based reference and DLS, and models without smoking status.** (a) We examined the ability of discrimination using C-statistic, reclassification improvement using category-free net reclassification improvement (cfNRI), and model calibration using the slope value from the reliability diagram. *In "Feature used" column, "Metadata" includes age, sex, and smoking status. (b) The sensitivity was calculated at the risk threshold matching specificity of the SBP-140 baseline at 63.7%, and the specificity was calculated based on the risk threshold matching sensitivity of the SBP-140 baseline at 55.2%. 95% confidence intervals (CIs) of C-statistic, cfNRI, and slope were obtained from the bootstrapping, and the p-values were computed by the permutation test. CIs of sensitivity and specificity were obtained from the Clopper-Pearson exact method, and the p-values were calculated by the permutation test with the prespecified margin of 2.5% and alpha of 0.05. The 95% CIs of NRI were computed by bootstrapping.
(DOCX)

**S1 Data.**
(DOCX)

## Acknowledgments

We acknowledge Nick Furlotte (Google Research) and the Google Research team for software infrastructure support. We thank Boris Babenko (Google Research) for his critical feedback on the manuscript. We also thank Madhuram Jajoo for the development of the open-source mobile application for collecting PPG signals. This research was conducted with the UK Biobank resource application 65275.

## Author Contributions

**Conceptualization:** Wei-Hung Weng, Sebastien Baur, Mayank Daswani, Sujay Kakarmath, Shruthi Prabhakara, Diego Ardila.

**Data curation:** Sebastien Baur, Mayank Daswani.

**Formal analysis:** Wei-Hung Weng, Mayank Daswani, Christina Chen, Lauren Harrell, Babak Behsaz, Cory Y. McLean, Yun Liu.

**Funding acquisition:** Yossi Matias, Greg S. Corrado, Shravya Shetty, Shruthi Prabhakara.

**Investigation:** Wei-Hung Weng, Sebastien Baur, Mayank Daswani, Sujay Kakarmath, Diego Ardila.

**Methodology:** Wei-Hung Weng, Sebastien Baur, Mayank Daswani, Mariam Jabara, Babak Behsaz, Cory Y. McLean, Yun Liu, Diego Ardila.

**Project administration:** Wei-Hung Weng, Sujay Kakarmath, Shruthi Prabhakara, Diego Ardila.

**Resources:** Yossi Matias, Greg S. Corrado, Shravya Shetty, Shruthi Prabhakara.

**Software:** Wei-Hung Weng, Sebastien Baur, Mayank Daswani, Mariam Jabara, Babak Behsaz, Cory Y. McLean.

**Supervision:** Wei-Hung Weng, Shruthi Prabhakara, Yun Liu, Goodarz Danaei, Diego Ardila.

**Validation:** Wei-Hung Weng, Sebastien Baur, Mayank Daswani, Sujay Kakarmath, Diego Ardila.

**Visualization:** Wei-Hung Weng, Yun Liu.

**Writing – original draft:** Wei-Hung Weng.

**Writing – review & editing:** Wei-Hung Weng, Sebastien Baur, Mayank Daswani, Christina Chen, Lauren Harrell, Sujay Kakarmath, Babak Behsaz, Cory Y. McLean, Yun Liu, Goodarz Danaei, Diego Ardila.

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
