## [Decision Letter · Decision Letter 0]

6 Oct 2023

PGPH-D-23-01000

Predicting cardiovascular disease risk using photoplethysmography and deep learning

Dear Dr. Weng,

Thank you for submitting your manuscript to PLOS Global Public Health. After careful consideration, we feel that it has merit but does not fully meet PLOS Global Public Health’s publication criteria as it currently stands. Therefore, we invite you to submit a revised version of the manuscript that addresses the points raised during the review process.

Please see the comments from one reviewer below. Please note that the reviewer comment about figure quality does not need attending to - this is an artifact of the PDF generation, and the original figures are of high enough quality.

Please note that we have only been able to secure a single reviewer to assess your manuscript. We are issuing a decision on your manuscript at this point to prevent further delays in the evaluation of your manuscript. Please be aware that the editor who handles your revised manuscript might find it necessary to invite additional reviewers to assess this work once the revised manuscript is submitted. However, we will aim to proceed on the basis of this single review if possible. 

We look forward to receiving your revised manuscript.

Kind regards,

Hanna Landenmark

Staff Editor

Journal Requirements:

Additional Editor Comments (if provided):

Reviewers' comments:

Reviewer's Responses to Questions

**Comments to the Author**

1. Does this manuscript meet PLOS Global Public Health’s publication criteria? Is the manuscript technically sound, and do the data support the conclusions? The manuscript must describe methodologically and ethically rigorous research with conclusions that are appropriately drawn based on the data presented.

Reviewer #1: Yes

2. Has the statistical analysis been performed appropriately and rigorously?

Reviewer #1: Yes

3. Have the authors made all data underlying the findings in their manuscript fully available (please refer to the Data Availability Statement at the start of the manuscript PDF file)?

Reviewer #1: Yes

4. Is the manuscript presented in an intelligible fashion and written in standard English?

Reviewer #1: Yes

5. Review Comments to the Author

Reviewer #1: In the introduction section results of the study are also provided in Figure 1.

The data is from UK biobank and hence how the results can be generalized to lower resource settings?

Cox proportional hazard was used to predict the ten-year risk of MACE among individuals free of CVD at baseline. Whether proportionality assumptions are satisfied? The predictors considered were using age, sex, smoking status and the results of analysis of PPG signals using deep learning.

The authors mentioned that the dataset was divided in to three train, tune and test based on latitude and longitude. But how this division was made based on latitude and longitude was not explained.

In Table 1 what is the meaning of “On hypertension medication, median [IQR]”. It is a count and not median. Similarly, for “On statin, median [IQR]”.

On an average how many follow-up visits were there for a participant?

What is the percentage of censoring in training, tune and test data used for the data analysis?

What was the criteria for selecting five Principal components? What is the total percentage of variance explained by these five components?

The DLS yielded C-statistic value is only 71.1% though is it significant, the C value is very low.

Figures are of poor clarity

6. PLOS authors have the option to publish the peer review history of their article (what does this mean?). If published, this will include your full peer review and any attached files.

**Do you want your identity to be public for this peer review?** For information about this choice, including consent withdrawal, please see our Privacy Policy.

Reviewer #1: No

---

## [Decision Letter · Decision Letter 1]

18 Dec 2023

PGPH-D-23-01000R1

Predicting cardiovascular disease risk using photoplethysmography and deep learning

Dear Dr. Weng,

Thank you for submitting your manuscript to PLOS Global Public Health. After careful consideration, we feel that it has merit but does not fully meet PLOS Global Public Health’s publication criteria as it currently stands. Therefore, we invite you to submit a revised version of the manuscript that addresses the points raised during the review process.

In addition to the reviewer’s comments, please address the following.

Include a detailed review of previous studies using photoplethysmography in cardiovascular risk assessment.Provide justification for the chosen deep learning model architecture instead of conventional analytical choice. Explain the data division process for training, validation, and testing.Show the comparison with existing cardiovascular risk scores with the approach chosen by the authors.Discussion can benefit from clinical implications given the study’s limitations. Also, please discuss potential future research directions in the conclusion.

We look forward to receiving your revised manuscript.

Kind regards,

Giridhara R Babu, MBBS, MPH, PhD

Academic Editor

Journal Requirements:

Additional Editor Comments (if provided):

Reviewers' comments:

Reviewer's Responses to Questions

**Comments to the Author**

1. If the authors have adequately addressed your comments raised in a previous round of review and you feel that this manuscript is now acceptable for publication, you may indicate that here to bypass the “Comments to the Author” section, enter your conflict of interest statement in the “Confidential to Editor” section, and submit your "Accept" recommendation.

Reviewer #1: All comments have been addressed

Reviewer #2: All comments have been addressed

Reviewer #3: (No Response)

2. Does this manuscript meet PLOS Global Public Health’s publication criteria? Is the manuscript technically sound, and do the data support the conclusions? The manuscript must describe methodologically and ethically rigorous research with conclusions that are appropriately drawn based on the data presented.

Reviewer #1: Yes

Reviewer #2: Yes

Reviewer #3: (No Response)

3. Has the statistical analysis been performed appropriately and rigorously?

Reviewer #1: Yes

Reviewer #2: Yes

Reviewer #3: (No Response)

4. Have the authors made all data underlying the findings in their manuscript fully available (please refer to the Data Availability Statement at the start of the manuscript PDF file)?

Reviewer #1: Yes

Reviewer #2: Yes

Reviewer #3: (No Response)

5. Is the manuscript presented in an intelligible fashion and written in standard English?

Reviewer #1: Yes

Reviewer #2: Yes

Reviewer #3: Yes

6. Review Comments to the Author

Reviewer #1: (No Response)

Reviewer #2: The authors analyzed PPG using deep learning and showed its usefulness using long-term tracking data on cardiovascular disease risk. This is a medically well-designed study. However, due to the lack of description of model development, the following items must be revised in order to make it a meaningful academic presentation.

1. A lot of information related to model development is missing from the paper. Describe in detail which patterns of PPG and which input format one-dimensional ResNet18 analyzed.

2. PPG is a time series signal. In general, performance improves by applying preprocessing related to signal processing or models related to time series analysis. Please describe whether any processing related to this was done.

3. Results must be shown regarding the interpretability of the deep learning model.

4. Signal measurement equipment such as PPG is subject to enormous bias depending on the equipment manufacturers. Describe UKB's PPG and other sets' equipment accurately and discuss bias.

Reviewer #3: 1. There is a major concern on the scientific basis underlying the results. It is known that the quality and morphology of PPG signals are highly sensitive to noises, and are dependent on many factors including wavelength, sensor attachment pressure, measurement site, and other physiological conditions, as well as the pre-processing method. The instability of PPG waveform is a major limitation in clinical application. In comparison, ECG morphology is more reliable. The proposed DLS contain important risk factors. The efficacy of PPG need to be evaluated independently.

2. Regarding the term "prediction", the time of follow-up is an important concern. The PPG morphology is age-dependent so age can be a confounding factor.

7. PLOS authors have the option to publish the peer review history of their article (what does this mean?). If published, this will include your full peer review and any attached files.

**Do you want your identity to be public for this peer review?** For information about this choice, including consent withdrawal, please see our Privacy Policy.

Reviewer #1: No

Reviewer #2: **Yes: **Tae Keun Yoo

Reviewer #3: No

---

## [Decision Letter · Decision Letter 2]

6 Feb 2024

PGPH-D-23-01000R2

Predicting cardiovascular disease risk using photoplethysmography and deep learning

Dear Dr. Weng,

Thank you for submitting your manuscript to PLOS Global Public Health. After careful consideration, we feel that it has merit but does not fully meet PLOS Global Public Health’s publication criteria as it currently stands. Therefore, we invite you to submit a revised version of the manuscript that addresses the points raised during the review process.

We look forward to receiving your revised manuscript.

Kind regards,

Jay-ar Formentera Medes

Support Staff - Editorial

Journal Requirements:

Additional Editor Comments (if provided):

Reviewers' comments:

Reviewer's Responses to Questions

**Comments to the Author**

1. If the authors have adequately addressed your comments raised in a previous round of review and you feel that this manuscript is now acceptable for publication, you may indicate that here to bypass the “Comments to the Author” section, enter your conflict of interest statement in the “Confidential to Editor” section, and submit your "Accept" recommendation.

Reviewer #1: All comments have been addressed

Reviewer #2: (No Response)

Reviewer #3: (No Response)

2. Does this manuscript meet PLOS Global Public Health’s publication criteria? Is the manuscript technically sound, and do the data support the conclusions? The manuscript must describe methodologically and ethically rigorous research with conclusions that are appropriately drawn based on the data presented.

Reviewer #1: Yes

Reviewer #2: Yes

Reviewer #3: (No Response)

3. Has the statistical analysis been performed appropriately and rigorously?

Reviewer #1: Yes

Reviewer #2: Yes

Reviewer #3: (No Response)

4. Have the authors made all data underlying the findings in their manuscript fully available (please refer to the Data Availability Statement at the start of the manuscript PDF file)?

Reviewer #1: Yes

Reviewer #2: Yes

Reviewer #3: (No Response)

5. Is the manuscript presented in an intelligible fashion and written in standard English?

Reviewer #1: Yes

Reviewer #2: Yes

Reviewer #3: (No Response)

6. Review Comments to the Author

Reviewer #1: The authors have addressed all the comments raised by reviewers

Reviewer #2: This study is expected to have high impact due to the big data analysis conducted by Google. For this reason, a more kind explanation is needed. Although there are coefficients for features in the supplementary material, the model in the presented paper is not explainable.

Regardless of the interpretation method, an interpretable explanation of which patterns are most closely related to cardiovascular disease is required along with representative images or sample signals of the analyzed PPG. Leaving the model developed in the paper as a blackbox reduces the value of the research. I suggest that authors present representative PPG figures that many researchers can understand.

Additionally, it is necessary to check throughout the paper whether supplementary materials are appropriately mentioned. There seem to be items that are numbered incorrectly or not mentioned.

Reviewer #3: (No Response)

7. PLOS authors have the option to publish the peer review history of their article (what does this mean?). If published, this will include your full peer review and any attached files.

**Do you want your identity to be public for this peer review?** For information about this choice, including consent withdrawal, please see our Privacy Policy.

Reviewer #1: No

Reviewer #2: **Yes: **Tae Keun Yoo

Reviewer #3: No

---

## [Decision Letter · Decision Letter 3]

16 Apr 2024

Predicting cardiovascular disease risk using photoplethysmography and deep learning

PGPH-D-23-01000R3

Dear Dr. Weng,

We are pleased to inform you that your manuscript 'Predicting cardiovascular disease risk using photoplethysmography and deep learning' has been provisionally accepted for publication in PLOS Global Public Health.

Best regards,

Julia Robinson

Executive Editor

Reviewer Comments (if any, and for reference):

Reviewer's Responses to Questions

**Comments to the Author**

1. If the authors have adequately addressed your comments raised in a previous round of review and you feel that this manuscript is now acceptable for publication, you may indicate that here to bypass the “Comments to the Author” section, enter your conflict of interest statement in the “Confidential to Editor” section, and submit your "Accept" recommendation.

Reviewer #2: All comments have been addressed

Reviewer #4: All comments have been addressed

2. Does this manuscript meet PLOS Global Public Health’s publication criteria? Is the manuscript technically sound, and do the data support the conclusions? The manuscript must describe methodologically and ethically rigorous research with conclusions that are appropriately drawn based on the data presented.

Reviewer #2: Yes

Reviewer #4: Yes

3. Has the statistical analysis been performed appropriately and rigorously?

Reviewer #2: Yes

Reviewer #4: Yes

4. Have the authors made all data underlying the findings in their manuscript fully available (please refer to the Data Availability Statement at the start of the manuscript PDF file)?

Reviewer #2: Yes

Reviewer #4: Yes

5. Is the manuscript presented in an intelligible fashion and written in standard English?

Reviewer #2: Yes

Reviewer #4: Yes

6. Review Comments to the Author

Reviewer #2: I would like to thank the authors for revising the paper well.

Reviewer #4: The manuscript is well thought out, explained rigorously and executed well.

7. PLOS authors have the option to publish the peer review history of their article (what does this mean?). If published, this will include your full peer review and any attached files.

**Do you want your identity to be public for this peer review?** For information about this choice, including consent withdrawal, please see our Privacy Policy.

Reviewer #2: **Yes: **Tae Keun Yoo

Reviewer #4: **Yes: **Dr Samaa Akhtar
